# Effect of Sample Preparation Pressure on Transformation Law of Low-Valent Titanium Oxide in a Multi-Stage Reduction Process

**Shigang Fan [1,2], Zhihe Dou [1,2,*], Ting'an Zhang [1,2], Ji-sen Yan [1,2] and Li-ping Niu [1,2]**

1   School of Metallurgy, Northeastern University, Shenyang 110819, China; fansg121212@163.com (S.F.); zta2000@163.net (T.Z.); 1810560@stu.neu.edu.cn (J.-s.Y.); niulp@smm.neu.edu.cn (L.-p.N.)

2   Key Laboratory of Ecological Metallurgy of Multi-metal Intergrown Ores of Ministry of Education, Shenyang 110819, China

*   Correspondence: douzh@smm.neu.edu.cn; Tel.: +86-13940091053

**Abstract:** A novel method for preparing titanium powder by multi-stage reduction of $TiO_2$ was proposed. Its core is the preparation of high-quality low-valent titanium oxide. In this paper, the effect mechanism of different sample preparation pressures on the preparation of low-valent titanium oxide by the primary reduction (self-propagating high-temperature synthesis mode, SHS) of the $Mg$-$TiO_2$ system was studied. The results show that the generation of Mg thermal fluid is the key link of the self-sustaining chemical reaction of the $Mg$-$TiO_2$ system. Titanium exists in $\alpha$-Ti and TiO at the end of combustion, and constitutes a non-stoichiometric low-valent titanium oxide. The sample preparation pressure determines the proportion of pores reserved for Mg diffusion in the compacts and the contact area of the reactants, thereby determining the partitioning behavior and heat transfer effect of Mg thermal fluid during the combustion process. When the sample preparation pressure is 75 MPa (relative density is $0.66 \pm 0.01$), the combustion effect is optimal, and the low-valent titanium oxide with oxygen content of 15.1% can be obtained. It was subjected to deep reduction to obtain a titanium powder product with an oxygen content of 0.27%.

**Keywords:** multi-stage reduction; thermal fluid; primary reduction; low-valent titanium oxide

## 1. Introduction

Titanium is one of the most important bulk commercial metals, and the current titanium industry is based on titanium sponge produced by the high-pollution, high-energy, non-continuous Kroll process [1]. The production cost of titanium sponge accounts for more than 1/3 of titanium alloy. A phase, β phase, α + β phase and composite titanium alloy materials have been developed, which are expected to be widely used in complex environments with harsh operating conditions [2–4]. Due to the requirements of the recycling economy and environmental protection, the development of low-cost, green titanium/titanium alloy technology has always been a hot topic. Avoiding high-temperature chlorination to prepare $TiCl_4$ intermediates, with a direct reduction in $TiO_2$ to prepare titanium, is one of the most promising new methods. Electrochemical metallurgy has the characteristics of a high energy utilization rate and high automation degree. Researchers have developed electrochemical reduction methods such as FFC (Derek J. Fray, Tom W. Farthing and George Zheng Chen) [5], OS (One and Suzuki) [6], EMR (electronically mediated reaction) [7], and SOM (solid oxide membrane) [8,9]. However, the technical problems of low current density and low current efficiency in the electrolytic reduction of $TiO_2$ have not been solved [5,10–12]. In order to obtain an electrode with stable electrical properties, the USTB (University of Science and Technology Beijing) method [13,14] was developed, that is, the $TiO_2$ was carbon-reduced and sintered into an electrically conductive Ti-C-O composite

intermediate as an anode to prepare titanium metal by electrolysis. Another type of low-cost clean preparation of titanium/titanium alloy processes is the metal thermal reduction process, that is, titanium/titanium alloy is prepared by high-temperature thermal reduction using the reactive metal (Mg, Ca, etc.) as the reducing agent and metal oxides such as $TiO_2$ as the raw material. Magnesium is cheap and readily available. It is almost insoluble in titanium and cannot form an intermetallic compound with titanium, and is an ideal reducing agent. However, the thermodynamic calculation results show that it is difficult to prepare titanium/titanium alloy with an oxygen content of less than 1% by magnesium thermal reduction of $TiO_2$ [15–17]. Calcium thermal reduction of $TiO_2$ can obtain titanium products with lower oxygen content. Based on this, the PRP (perform reduction process) method needs to add a large amount of $CaCl_2$ auxiliary agent to suppress side reactions. The method has the defects of poor kinetic conditions and the presence of $CaTiO_3$ compounds, resulting in low reduction efficiency [18,19]. In the presence of additives, there are reports that excess Mg directly reduces $TiO_2$ (mole ratio of $Mg/TiO_2$ is greater than 2.0) [20–22].

Based on the thermodynamic evolution of $TiO_2$ reduction and the chemical potential of different reducing agents, a new idea of multi-stage deep reduction to prepare titanium/titanium alloy was proposed [23], First, $TiO_2$ is subjected to magnesium thermal reduction (self-propagating high-temperature synthesis mode, SHS mode) to obtain the non-stoichiometric low-valent titanium oxide containing MgO by-product, that is, primary reduction; then, the primary reduction product is subjected to calcium thermal reduction to obtain the deep reduction product containing CaO by-product, that is, deep reduction. Finally, the deep reduction product is subjected to enhanced acid leaching, filtration, and drying to obtain titanium powder. "First Mg and then Ca" can fully utilize chemical energy or avoid the formation of a compound phase in which $CaTiO_3$ is difficult to reduce. The process is based on the formation and reduction of low-valent titanium oxide. The SHS reaction form can be used to rapidly prepare highly active mesophases. At the same time, it has changed the current situation of high pollution and high energy consumption in the preparation of intermediate phases such as $TiCl_4$ in the titanium industry. Therefore, the core of the process is the preparation of high activity low-valent titanium oxide intermediates.

As a special powder metallurgy method, sample preparation pressure is an important process parameter of SHS that has a decisive influence on the reaction behavior and reaction effect [24,25]. The sample preparation pressure determines the contact of the reactants, as well as the green density (relative density), which in turn affects the thermal conductivity behavior in the reaction. Sample preparation pressure is used as a kinetic parameter, and its effect on different combustion systems has been reported [25–28]. In different combustion systems, the sample preparation pressure will determine the process parameters such as porosity and relative density of the compact, which will have an important impact on the combustion process and conversion behavior of the reactants [26–36]. In this paper, the Mg-$TiO_2$ SHS system (primary reduction reaction) was studied, and the research was focused on the effect of sample preparation pressure on the primary reduction process and the evolution of low-valent titanium oxide.

## 2. Materials and Methods

### 2.1. Experimental Procedures

Experimental materials: rutile titanium dioxide ($TiO_2 > 98.50\%$); magnesium powder (Mg > 99.00%, 38–75 μm); leaching agent—hydrochloric acid (HCl > 36.00%).

Experimental procedure: First, $TiO_2$ was dried at 110 °C for 24 h; then, it was mixed with Mg powder in proportion (molar ratio Mg:$TiO_2$ = 2:1), and pressed into Φ35 mm compacts with different sample preparation pressures. The compacts were placed in a reaction crucible and subjected to SHS reaction to obtain the primary reduction product. SHS reaction was initiated in a local ignition mode. The prepared compacts and crucibles were placed in a sealed reactor, evacuated to 100 Pa, filled with high purity argon (>99.999%) to atmospheric pressure, and repeated three times. Finally, the compacts

were held in 1 atm Ar, and the heating wire was energized to initiate SHS reaction from one end of the compact until the end of combustion to obtain a primary reduction product. The primary reduction product was leached in dilute hydrochloric acid (1 mol/L) to remove the MgO by-product, and after filtration and drying, the low-valent titanium oxide intermediate having a non-stoichiometric composition was obtained, which can be used as raw material for multi-stage deep reduction to prepare titanium/titanium alloy.

The rutile $TiO_2$ (Tetragonal) used in the experiment is the nano-powder (as shown in Figure 1), and its particle size distribution is in the narrow range of 100–600 nm. The Mg particle size is 38–75 μm. The sample preparation pressure of the reactants was controlled to 25, 50, 75, 100, and 150 MPa, respectively. After 20 min, the whole compact was taken from the mold of the press for subsequent experiments and characterization. The specific experimental scheme is shown in Table 1.

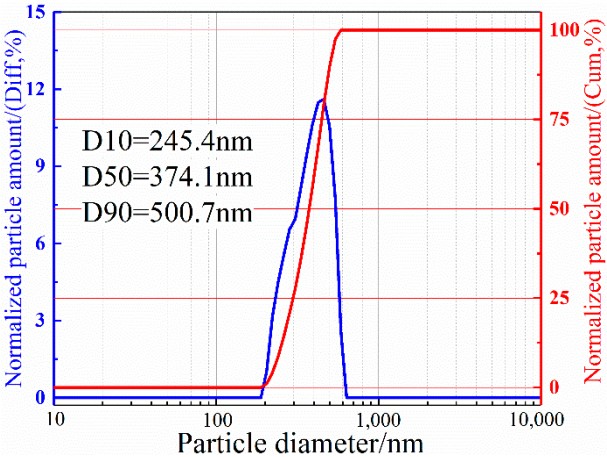

**Figure 1.** $TiO_2$ raw material particle size distribution.

**Table 1.** Self-propagating high-temperature synthesis mode (SHS) reaction conditions.

| No. | Sample Preparation Pressure/MPa | Mole Ratio of $TiO_2$/Mg |
|---|---|---|
| 1 | 25 | 1:2 |
| 2 | 50 | 1:2 |
| 3 | 75 | 1:2 |
| 4 | 100 | 1:2 |
| 5 | 150 | 1:2 |

### 2.2. Analysis and Characterization

The product phase was analyzed by D8 ADVANCE X-ray diffractometer (Cu target, Kα, 4°/min, Bruker Corporation, Billerica, MA, USA); the microstructure and composition analysis of the reaction product were measured by field emission with a scanning electron microscope (SU8010, HITACHI, Tokyo, Japan) and energy dispersive X-ray spectroscopy (Bruker,20 mA, 20 KV, Bruker Corporation, Billerica, MA, USA); the oxygen content of the primary reduction product(leached) was analyzed by an oxygen, nitrogen, hydrogen analyzer (LECO ONH836, LECO, St. Joseph, IN, USA); the particle size distribution characterization of the $TiO_2$ raw material and primary reduction product (leached) was carried out with a Mastersize 3000 laser particle size analyzer (Malvern Panalytical, Worcestershire, UK).

The prepared compact was weighed in mass and volume, and the actual density was calculated. At the experimental conditions (mole ratio of Mg: $TiO_2$ = 2:1), we can calculate the relative density of the compact, that is, the ratio of the actual density to the theoretical density (fully dense compact). Three compacts were pressed under the same sample pressure to determine the average density.

## 3. Results and Discussion

### 3.1. Phase Analyses

As shown in Figure 2, when the sample preparation pressure is 25 MPa (Figure 2a), the product is the low-valent titanium oxide phase represented by TiO, $\alpha$-Ti and the MgO by-product, while a small amount of unreacted Mg exists; when the sample preparation pressure is increased to 50–75 MPa (Figure 2b,c), the titanium-containing phases of the product are TiO and $\alpha$-Ti, and the diffraction peak of $\alpha$-Ti phase tends to decrease, while Mg phase disappears; when the sample preparation pressure is increased to 100 MPa (Figure 2d), Mg phase appears in the product; and when the sample preparation pressure is increased to 150 MPa (Figure 2e), the diffraction peak of $\alpha$-Ti phase in the product is remarkably lowered.

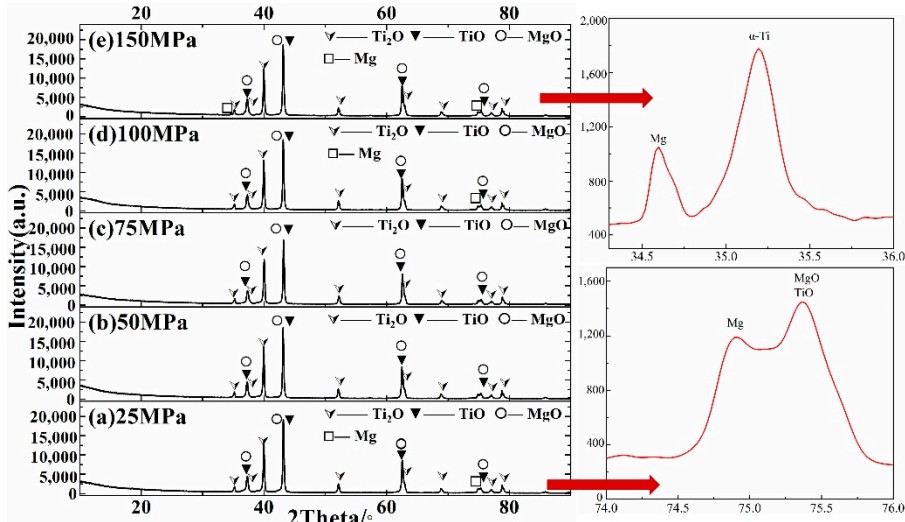

**Figure 2.** XRD patterns for SHS products of the Mg-TiO$_2$ system at different sample preparation pressures.

The XRD patterns in Figure 3 show the phase change of the titanium-containing component phase in the combustion products of Mg-TiO$_2$ system at different sample preparation pressures after removal of Mg and MgO with hydrochloric acid. Two titanium-containing phases are present; $\alpha$-Ti and TiO. With the increase in sample preparation pressure, the height of the diffraction peak first decreases and then increases. When the sample preparation pressure is 75 MPa (Figure 3c), the diffraction peak height is the lowest. This is because the contact area and porosity of TiO$_2$ are suitable at this time, and the space for diffusion of Mg and the path of high-temperature conduction are better matched. As a consequence, the combustion speed is fast, the time for crystal lattice reorganization of the titanium-containing phase is relatively short, and the crystal development is relatively incomplete. $\alpha$-Ti has an HCP (Hexagonal Close-Packed) structure. The XRD curve obtained in Figure 3c is processed by Jade 7.0 software, and its lattice constant can be obtained as a = b = 2.9613 Å and c = 4.8320 Å. TiO has a BCC (Body-Centered Cubic) structure, and its crystal lattice constant is a = b = c = 4.1722 Å. As the sample preparation pressure is increased to 150 MPa (Figure 3e), the diffraction peak of $\alpha$-Ti phase in the product is significantly reduced. It indicates that the porosity of the material particles becomes small due to the excessive sample preparation pressure, and Mg diffusion in the compact is limited, resulting in insufficient an combustion reaction. The titanium-containing phase cannot be further converted from TiO phase to $\alpha$-Ti phase.

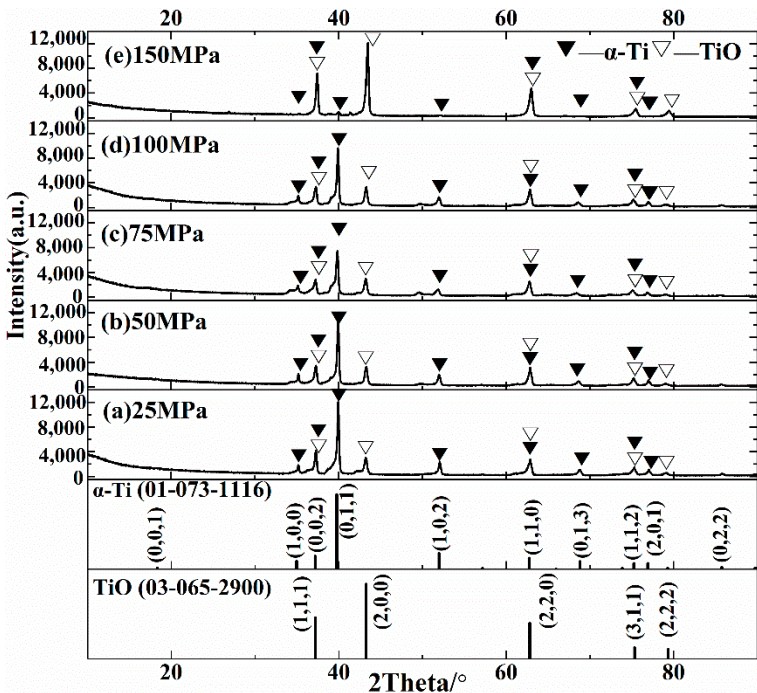

**Figure 3.** XRD patterns for SHS products (leached) of the Mg-TiO$_2$ system at different sample preparation pressures.

## 3.2. Micromorphology Analysis

Figure 4 shows the microscopic morphology of combustion products of the Mg-TiO$_2$ system at different sample preparation pressures. When the sample preparation pressure is 25 MPa, the product morphology is a dense block (red region in Figure 4a) and loose small particle accumulation (blue region in Figure 4a). According to the previous analysis, this is due to the difference in the relative positions of Mg and TiO$_2$, and Mg participates in the combustion reaction at differently condensed phases. The composition of the combustion product particles is Mg-MgO-Ti$_x$O (0 < x < 1), that is, the low-valent titanium oxide phase wrapped by a Mg-MgO layer. According to the results of EDS analysis presented in Table 2, the dense zone (points A and B in Table 2) has more Mg content than the loose zone (points C and D in Table 2), which indicates that the dense zone has a flow of excess Mg through the pores of the reactant skeleton, which is subsequently condensed. As the sample preparation pressure increases to 75 MPa, the product particles become looser (compare the red and blue regions in Figure 4a,b), and the particles are still Mg-MgO-wrapped low-valent titanium oxide (points E, F, G, and H in Table 2). When the sample preparation pressure is increased to 150 MPa, the surface of the product particles becomes dense (red region in Figure 4c), the particle size of the loose zone is significantly enlarged (blue region in Figure 4c), and Mg in the product particles is significantly higher (points I, J in Table 2). The dense zone shows the state in which the excess fluid Mg is solidified (point I in Figure 5c), indicating that the porosity among TiO$_2$ particles is reduced due to the increase in sample preparation pressure. After the reaction of TiO$_2$ particles in contact with Mg, the resulting liquid or gaseous Mg lack diffused channels, and the phenomenon of the red region in Figure 4c is generated; due to the tighter contact between TiO$_2$ particles, the liquid or gaseous Mg is poorly contacted with the individual TiO$_2$ particles, while a group of particles is formed by coating a plurality of TiO$_2$ particles. This forms the phenomenon of coarse monomer particles in the blue region of Figure 4c.

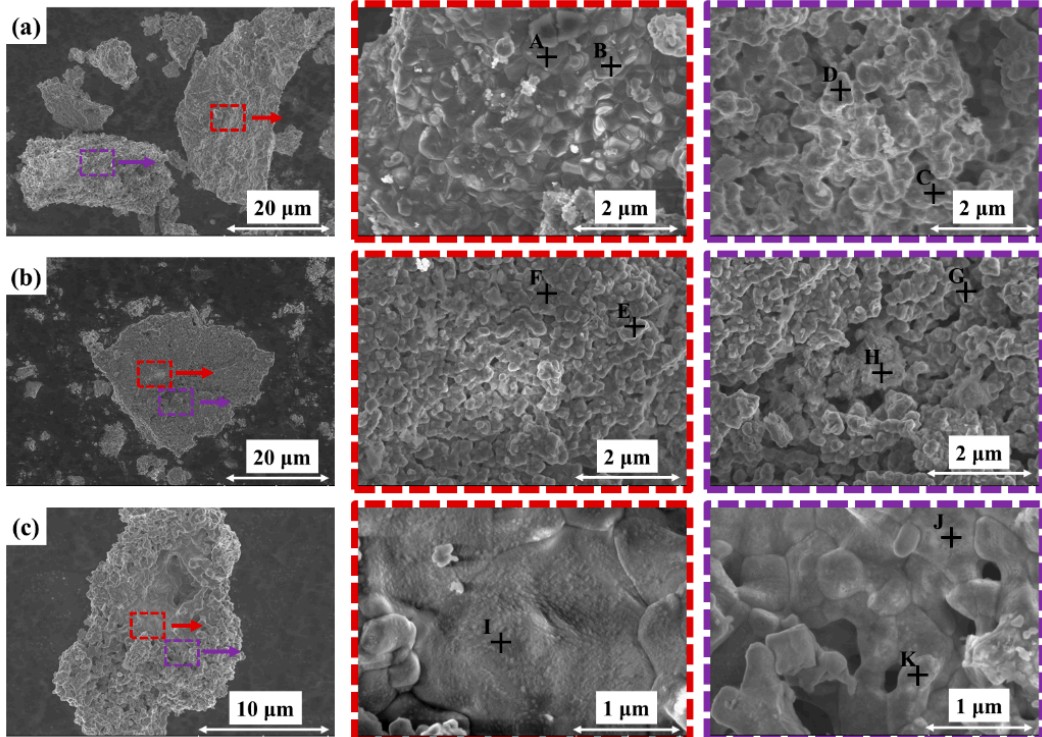

**Figure 4.** SEM photos for SHS products of the Mg-TiO$_2$ system at different sample preparation pressures: (**a**) 25; (**b**) 75; (**c**) 150 MPa.

**Table 2.** EDS results of SHS products at different sample preparation pressures.

| No. | Ti/(wt.%) | O/(wt.%) | Mg/(wt.%) | Atomic Ratio of Ti/O/Mg | |
| --- | --- | --- | --- | --- | --- |
| A | 38.31 | 26.63 | 35.06 | | 1:2.1:1.8 |
| B | 34.40 | 30.07 | 35.53 | | 1:2.6:2.0 |
| C | 49.16 | 18.84 | 32.00 | | 1:1.2:1.3 |
| D | 49.71 | 20.87 | 29.42 | | 1:1.3:1.2 |
| E | 34.80 | 29.81 | 35.39 | | 1:2.6:2.0 |
| F | 39.86 | 22.66 | 37.48 | | 1:1.7:1.9 |
| G | 42.74 | 29.07 | 28.19 | | 1:2.0:1.3 |
| H | 53.13 | 29.71 | 17.16 | | 1:1.7:0.6 |
| I | 47.88 | 20.05 | 32.08 | | 1:1.3:1.3 |
| J | 50.85 | 21.47 | 27.69 | | 1:1.3:1.1 |
| K | 38.27 | 31.33 | 30.40 | | 1:2.5:1.6 |

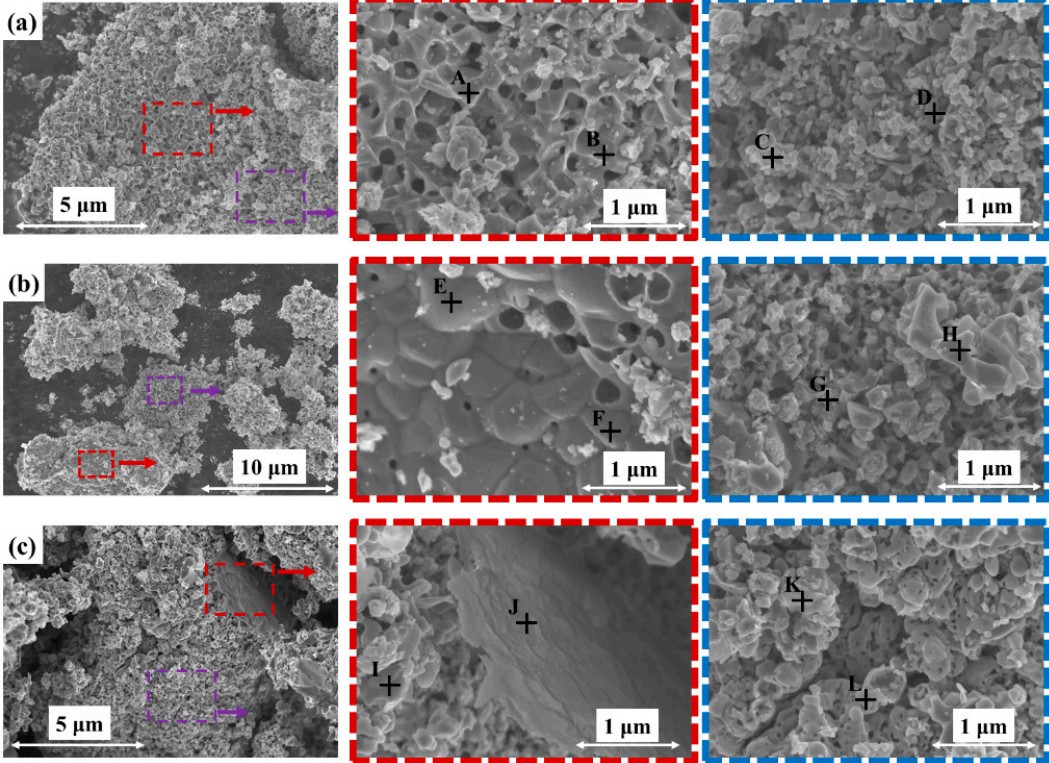

**Figure 5.** SEM photos for SHS products(leached) of the Mg-TiO$_2$ system at different sample preparation pressures: (**a**) 25; (**b**) 75; (**c**) 150 MPa.

The self-sustained combustion reaction process of the Mg-TiO$_2$ system can be summarized as the combustion zone in the initial reaction to transfer the local excess Mg to the unreacted region, continue to burn on the surface of TiO$_2$ to generate high temperature, and further transfer energy (high temperature) and mass (excess Mg). Each combustion zone reacts as a "transfer station" to transfer excess Mg to the unreacted zone, so that Mg and TiO$_2$ separated from each other before the reaction form Mg-MgO-wrapped low-valent titanium oxide particles. When the sample preparation pressure is small (25 MPa), the distance between TiO$_2$ particles is far, and the excess Mg is difficult to be transported to the unreacted TiO$_2$ region, that is, the shortage of the "transfer station" is difficult to generate the energy required for transporting Mg, resulting in excess Mg condensation and forming a dense zone in the red region of Figure 4a. The sample preparation pressure is further increased, and when the contact area among the TiO$_2$ particles and the porosity are properly matched, Mg can be better distributed and reacted in the combustion system. The combustion products generated at this time are loose, uniform small particle accumulations, as shown in Figure 4b. When the sample preparation pressure is too large (150 MPa), the porosity among TiO$_2$ particles is too low. This makes excess Mg difficult to pass through the combustion zone. Mg that is partially excessive and fails to diffuse in the red region of Figure 4c is formed.

Figure 5 shows the microstructure of titanium-containing components in the combustion products of the Mg-TiO$_2$ system at different sample preparation pressures after removal of Mg and MgO with hydrochloric acid. When the sample preparation pressure is 25 MPa, the product is an open-cell skeleton in which particles are stacked at a size of 1 μm (red region in Figure 5a) and 200–400 nm (blue region in Figure 5a). This diversity is due to the difference in the relative positions of the TiO$_2$ particles with Mg during the combustion process, resulting in a difference in the order of contact with high temperature (sintering) or with Mg (reduction). As the sample preparation pressure is increased to 75 MPa, low-valent titanium oxide is sintered during the reaction (red region in Figure 5b), and the particle size is significantly enlarged. According to EDS analysis results in Table 3, the oxygen contents of points A

and B are higher than those of points E and F, which is caused by the sintering of the titanium-containing compound increasing the distance of the oxygen atoms diffuse. Low-valent titanium oxide particles larger than 1 μm appear (point H in Figure 5b). This is due to the excessive sample preparation pressure causing the agglomerated $TiO_2$ particles to contact more closely, forming particle groups. The titanium oxide is solid in the whole reaction process, and the liquid or gaseous Mg diffuses to the surface to form a "floating island", and Mg can only react on the surface of the $TiO_2$ particle groups and cannot react with each $TiO_2$ particle alone. This also enlarges the distance of oxygen diffusion, leading to the advancement of the combustion end point, and, therefore, the oxygen content of the large particles is high (points G, H in Figure 5b and EDS analysis results in Table 3). When the sample preparation pressure continues to increase to 150 MPa, the open-cell skeleton in the dense zone is transformed into a closed-cell structure (red region in Figure 5a–c). At the same time, Mg content in the closed-cell region is high (point J in Figure 5c and EDS analysis result in Table 3). This is due to the reduced contact probability of the MgO by-product in the closed-cell region with $H^+$ during acid leaching. At the same time, comparing the loose zone in the blue regions of Figure 5a–c, the size of the monomer particles constituting the skeleton structure also increases. This is also caused by the rapid "sintering" phenomenon between the titanium oxide particles.

**Table 3.** EDS results of SHS products (leached) at different sample preparation pressures.

| No. | Ti/(wt.%) | O/(wt.%) | Mg/(wt.%) |
|-----|-----------|----------|-----------|
| A | 88.31 | 11.22 | 0.47 |
| B | 86.20 | 13.80 | 0.00 |
| C | 87.39 | 12.07 | 0.54 |
| D | 85.90 | 13.75 | 0.35 |
| E | 77.32 | 21.99 | 0.69 |
| F | 74.14 | 25.22 | 0.64 |
| G | 89.53 | 9.58 | 0.89 |
| H | 73.70 | 25.55 | 0.75 |
| I | 71.28 | 27.61 | 0.71 |
| J | 81.28 | 15.43 | 3.29 |
| K | 84.97 | 14.42 | 0.71 |
| L | 88.13 | 11.61 | 0.26 |

*3.3. Action Mechanism of Sample Preparation Pressure on Combustion Reaction*

As shown in Figure 6, as the pressure applied to $Mg-TiO_2$ compact increases, the relative density of the compact increases. The oxygen content in the product tends to decrease first and then increases, and the minimum value occurs when the sample preparation pressure is 75 MPa. In terms of phase composition, α-Ti phase and TiO phase simultaneously exhibit maximum and minimum values at 75 MPa. As the sample preparation pressure increases, the contact area of the reactants in the compact increases, and the porosity decreases. The former facilitates the combustion reaction, while the latter is unfavorable for the Mg diffusion of the process to continue to participate in the reaction. At the condition that the relative content and the particle size of the raw materials (Mg and $TiO_2$) are the same, the proportion of the pores reserved for Mg diffusion in the compact and the contact area of the reactants can be determined by adjusting the sample preparation pressure in order to achieve the purpose of controlling the combustion reaction progress. At the condition of a sample preparation pressure of 25/75/150 MPa, using the Pt-30 Rh type thermocouple to measure the burning speed of the compact with a length of ~40 mm, they are 2.4/4.1/2.9 mm/s, respectively. This indicates that the sample preparation pressure affects the diffusion behavior of Mg, which in turn affects the combustion speed.

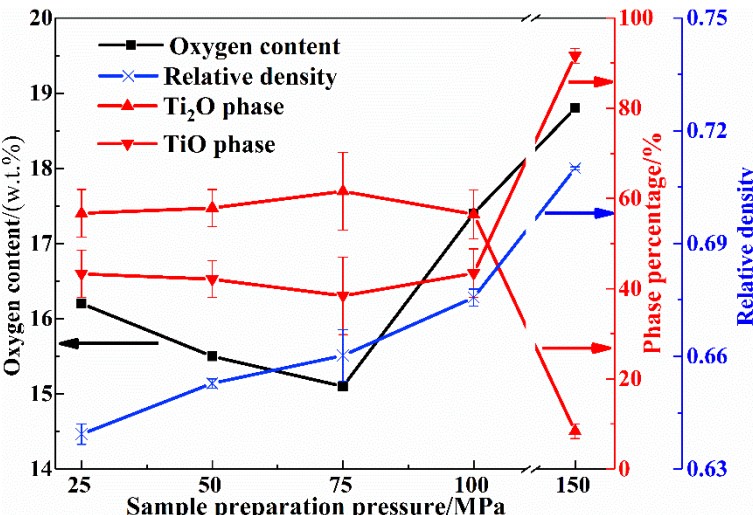

**Figure 6.** Oxygen content/phase composition/relative density of combustion products (leached) of the Mg-TiO$_2$ system at different sample preparation pressures.

SHS reaction is a complex multiphase, fast combustion reaction. Adiabatic temperature is the most important thermochemical parameter for studying self-sustaining chemical reactions [24,37]. The basic assumptions of the adiabatic temperature are: (1) complete conversion of each material in the reaction to produce the target product; (2) the combustion reaction occurs and completes instantaneously; and (3) the heat generated by the reaction is completely used to heat the product without heat exchange with the outside. When the above three points are met, the highest temperature of the system is the adiabatic temperature. However, assumption (1) ignores the thermodynamic limitations of the equilibrium of the system reaction, and assumption (2) ignores the kinetic limitations of the combustion reaction rate. Classical adiabatic temperatures resulting from the above assumptions do not accurately describe all reaction systems. According to the phase analysis in Figures 2 and 3 and the composition analysis in Figure 6, the SHS product of the Mg-TiO$_2$ system can only obtain low-valent titanium oxide between Ti and TiO, instead of Ti. This indicates that the combustion degree of this system is incomplete, and it is a reaction system limited by thermodynamic limits. The literature [23,26,29,38–43] also shows that due to the incomplete conversion of materials in the titanium-containing system, a difference between the actual reaction temperature and the classical adiabatic temperature calculation will occur.

Therefore, according to the reduction degree of the Mg-TiO$_2$ system, the adiabatic temperature of the reaction to form Ti (2Mg + TiO$_2$ = Ti + 2MgO) and TiO (Mg + TiO$_2$ = TiO + 2MgO) is calculated as the upper and lower limits of the actual combustion temperature, and the combustion temperature range at different ratios can be obtained (Figure 7). The upper and lower limits of the reaction include the actual adiabatic temperature at which the titanium-containing compound is formed between Ti and TiO (broad solid solution region of oxygen in the Ti-O binary phase diagram).

According to the calculation, the upper limit of the adiabatic temperature of the Mg-TiO$_2$ system increases first and then decreases with the increase in Mg ratio, and traverses the melting points of Ti (Mg: TiO$_2$ = 1.19–1.31) and TiO (Mg: TiO$_2$ = 1.76–1.92). When the ratio is the theoretical equivalent (Mg: TiO$_2$ = 2.0), the highest temperature (2067.6 K) is reached. As the Mg ratio is further increased, the adiabatic temperature decreases (excess Mg cannot participate further in the reaction and actually acts as a diluent), traversing the melting point of titanium (Mg: TiO$_2$ = 2.11–2.21) and the boiling point of Mg (Mg: TiO$_2$ = 2.70–3.00). The lower limit of the adiabatic temperature of the Mg-TiO$_2$ system decreases with the increase in the Mg ratio, stays at the boiling point of Mg within a certain ratio (Mg: TiO$_2$ = 1.37–2.50), and further decreases to the phase transition temperature of TiO (Mg: TiO$_2$ = 2.94–3.00). According to the analysis, the adiabatic temperature of the Mg-TiO$_2$ system is greatly

affected by thermodynamic constraints of the reaction process. In the range of Mg: $TiO_2$ = 1.00–2.70, the actual adiabatic temperature is in the range of 2067.6–1363.0 K; in the range of Mg: $TiO_2$ = 2.70–3.00, the actual adiabatic temperature is in the narrow range of 1363.0–1264.0 K.

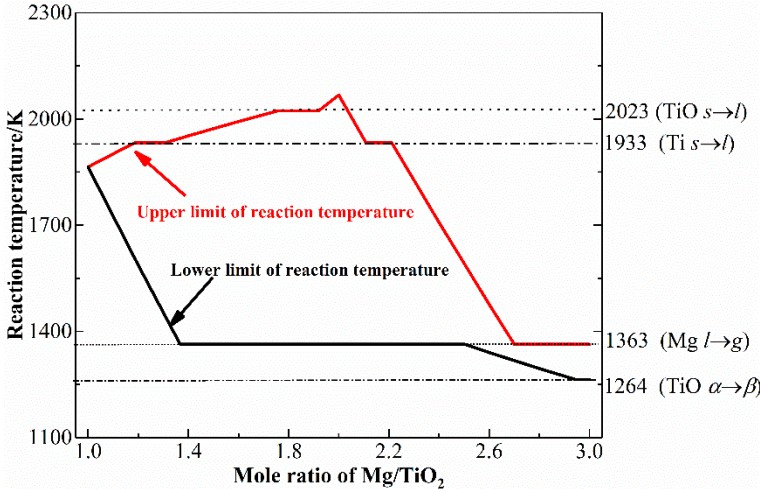

**Figure 7.** Actual combustion temperature range of the Mg-$TiO_2$ system (Initiation temperature 298 K).

The actual combustion temperature range is in the range of 1363–2076.6 K at the experimental conditions (Mg: $TiO_2$ = 2.0; initiation temperature: 298 K). According to the phase analysis in Figure 3 and the composition analysis in Figure 6, the product is low-valent titanium oxide composed of TiO and $\alpha$-Ti phase (solid solution of oxygen in titanium), so the actual combustion temperature is closer to the lower limit of the calculated temperature. At this time, the titanium-containing phase and the continuously formed by-product MgO maintain a solid phase throughout the combustion process, and a local excess of Mg undergoes liquefaction or even gasification to participate in the reaction. The kinetics of the Mg-$TiO_2$ system was analyzed, and the reaction initiation temperature is 530–607 °C [44–46], and the solid–solid reaction is carried out at this temperature. In summary, we can obtain the combustion reaction mode of the Mg-$TiO_2$ system as:

(1) Initial stage: the material in the heating wire heating zone reaches the reaction initiation temperature, and the combustion reaction starts. Because the reaction mode is a solid–solid reaction, the reaction can only occur at the surface of $TiO_2$ which is in direct contact with Mg;

(2) Flow phases generation stage: The elevated temperature generated in the previous stage causes local excess Mg liquefaction and gasification (calculation result in Figure 7), and the generated fluid Mg diffuses through the pores of the particles under the action of surface tension and gravity (compact was ignited from the upper end);

(3) Spreading stage: The thermal fluid generated in the previous stage diffuses to the surface of $TiO_2$ which is not in direct contact with the solid Mg and undergoes forced heat exchange, and the combustion continues when the temperature reaches the combustion initiation temperature.

The space expansion of the combustion wave is achieved by the cycle of stages (1) to (3) until the end of the reaction. For the combustion reaction of such a solid–solid reaction system, since the initial reaction interface is the contact portion of the solid particles, it is impossible to contain all the atoms in the system, so the fluidized intermediate phase generation is an inevitable need for the self-sustaining reaction.

The sample preparation pressure directly determines the contact area of the reaction material and the porosity of the compact. According to the analysis of the reaction process, Mg thermal fluid will reach a new reaction interface by diffusion to initiate a new combustion reaction in order to achieve the purpose of self-sustaining reaction. Therefore, the above two conditions have an influence on the

generation and diffusion process of Mg thermal fluid, which in turn determines the progress of the combustion reaction. When the sample preparation pressure is small, the porosity of the material is large, and the diffusion resistance of Mg thermal fluid is small, but the contact area of the material is small, and the surface tension is relatively small, which is not conducive to the combustion reaction; when the sample preparation pressure is too large, the material is in close contact. It is beneficial to heat transfer (combustion wave), as it allows for the pores in the compact to be more finely dispersed and provides a large surface tension. However, when the porosity is too small, the resistance of Mg fluid diffusion is too large, and the excessive Mg may not reach the unreacted $TiO_2$ surface and remain in the compact (red region in Figure 4c). When the sample preparation pressure is too large, $TiO_2$ particles in the compact are too tightly contacted, and it is easy to form an excessively low-valent titanium oxide agglomerate in the elevated temperature generated by combustion (as shown in the red region in Figure 5b,c). This prolongs the diffusion of oxygen in the particles and deteriorates the reduction effect. We regard $TiO_2$ particles in direct contact with Mg as the starting zone of the combustion reaction—the region where forced heat exchange with Mg thermal fluid is used to initiate the combustion reaction as a development zone. It can be seen that the development zone is both an inevitable result of starting zone and an inevitable need of a self-sustaining combustion system. In fact, the development zone can be seen as a transit station for combustion behavior, which consumes excess Mg fluid in the starting zone, as well as further elevated temperature and fresh Mg fluid to initiate a chain reaction. Therefore, a suitable sample preparation pressure will bring about a reasonable porosity and reaction interface area, which can optimally distribute Mg fluid and the reaction-elevated temperature. At the same time, the sample preparation pressure determines the initial state and spatial layout of $TiO_2$ particles before the reaction, which affects the agglomeration behavior and particle size of the titanium-containing compound in the combustion reaction. Excessive sample preparation pressure causes $TiO_2$ particles to be too tightly contacted, and it is easy to form a closed-cell structure (red region in Figure 5c). Based on the above analysis, a relatively suitable sample preparation pressure for combustion is 75 MPa, and the relative density is 0.66 ± 0.01 at this time.

According to the above description, by initiating the SHS reaction of the Mg-$TiO_2$ system, the non-stoichiometric low-valent titanium oxide with an oxygen content of 15–20% can be obtained. The sample preparation pressure has a decisive influence on the phase and morphology of the low-valent titanium oxide. According to Figures 5b and 6, this intermediate is a fine pore structure with an oxygen content of 15.1% (sample preparation pressure: 75 MPa). It was used as a raw material, and calcium was added to a deep deoxidation reduction at 1000 °C for 2 h, and the product was leached with 1 mol/L dilute hydrochloric acid to obtain the deep reduction product shown in Figure 8. The oxygen content in the product is only 0.27%, and the microscopic morphology is an irregular block. The monomer particle size is several hundred nanometers to 2 μm (Figure 8a). The whole particle has a porous skeleton shape (Figure 8a), which inherits the microscopic morphology of the low-valent titanium oxide (Figure 5b). Its particle size increases, indicating that stable elevated temperature reduction leads to sintering of low-valent titanium oxide particles.

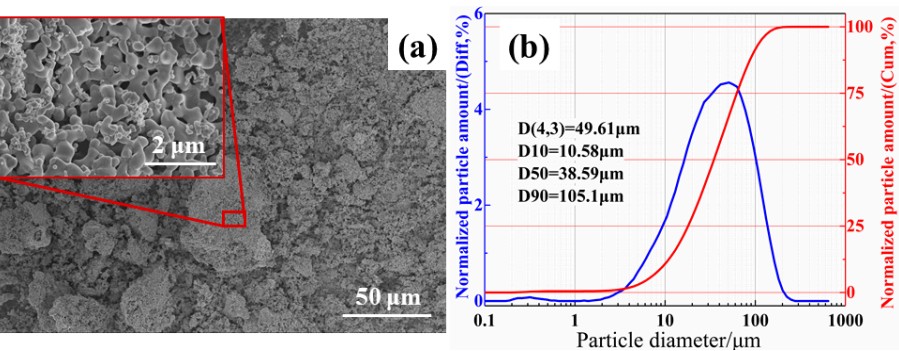

**Figure 8.** SEM photo (**a**) and particle size distribution (**b**) of the deep reduction product (leached).

## 4. Conclusions

In this paper, SHS of Mg-TiO$_2$ was studied, and the effects of different sample preparation pressures on the combustion behavior and phase transformation of the product were studied. The following conclusions were obtained:

(1) The generation of Mg thermal fluid is the key link of self-sustaining chemical reaction. When combustion is terminated, titanium exists in TiO and α-Ti phase, and constitutes a non-stoichiometric low-valent titanium oxide;

(2) The titanium-containing phase exists in a solid state and simultaneous sintering and reduction reactions occur during the combustion reaction, and a skeleton form composed of low-valent titanium oxide particles is finally exhibited. Excessive sample preparation pressure causes the titanium-containing compound to change from an open cell structure to a closed cell structure;

(3) The sample preparation pressure determines the proportion of pores reserved for Mg diffusion in the compacts and the contact area of the reactants, thereby affecting the partitioning behavior of Mg thermal fluid and the heat transfer effect during the combustion process. When the sample preparation pressure is 75 MPa (relative density 0.66 ± 0.01), the combustion effect is optimal, and a low-valent titanium oxide precursor with an oxygen content of 15.1% can be obtained. After the deep reduction of low-valent titanium oxide precursor, a titanium powder product with oxygen content of 0.27% can be obtained.

**Author Contributions:** Determined the theme of the article and undertook the experiment, characterization and writing work, S.F.; reviewed the full text and provided experimental platform support, Z.D. and T.Z.; provided assistance in data processing, J.-s.Y. and L.-p.N. All authors have read and agreed to the published version of the manuscript.

**Funding:** This research was supported by the National Natural Science Foundation of China (U1908225, U1702253, 51774078), the Fundamental Research Funds for the Central Universities (N172506009, N170908001, N182515007).

**Conflicts of Interest:** The authors declare that they have no known competing financial interests or personal relationships that could have appeared to influence the work reported in this paper.

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
