# Peer review of "Effect of Sample Preparation Pressure on Transformation Law of Low-Valent Titanium Oxide in a Multi-Stage Reduction Process"

_metals, doi:10.3390/met10091259_

Round 1

Reviewer 1 Report

Please check the text thoroughly. The text still contains parts of the journal instructions (eg lines 79-84, 150-151). Furthermore, although the language in the majority of the manuscript is appropriate, there are certain cases where language improvement is needed.

Please give the full name of certain processes (eg PRP, etc.) before giving their abbreviation

It is proposed that the paper is accepted for publication

Author Response

Thank you for reviewing my manuscript. My reply to your comment is as follows:

Q1. Please check the text thoroughly. The text still contains parts of the journal instructions (eg lines 79-84, 150-151). Furthermore, although the language in the majority of the manuscript is appropriate, there are certain cases where language improvement is needed.

A1:Thank you for your valuable comments. I have carefully reviewed the full article and deleted the parts of the journal instructions (eg lines 79-84, 150-151). The language of the full article was revised based on the review opinions of colleagues with experience studying in a country where English is the native language.

Q2:Please give the full name of certain processes (eg PRP, etc.) before giving their abbreviation。

A2: Thank you for your valuable comments. I have reviewed the full article. The abbreviation that appears for the first time in the article is explained and supplemented. (eg lines 37-38,41,51)

Q3: It is proposed that the paper is accepted for publication

A3: Thank you for your valuable comments. I have reviewed and revised the full article.

Reviewer 2 Report

Overall, this is a very solid paper. Major revisions are in order as there are some points that need to be clarified by the authors.

English needs to be revised by a native speaker. Several grammatical mistakes are found in the text.

On the importance of Ti and its alloys refer to different potential applications as exemplified in the following papers: 10.1016/j.actamat.2015.12.021, 10.1016/j.matchar.2020.110400 and 10.1016/j.matchar.2020.110180.

“FFC[2], OS[3], EMR[4], SOM[5, 6].”: write in full the names of these processes.

“Materials and Methods should be described with sufficient details to allow others to replicate 79 and build on published results. Please note that publication of your manuscript implicates that you 80 must make all materials, data, computer code, and protocols associated with the publication available 81 to readers. Please disclose at the submission stage any restrictions on the availability of materials or 82 information. New methods and protocols should be described in detail while well-established 83 methods can be briefly described and appropriately cited.”: this should be removed from the paper.

What was the grain size of TiO2?

“preparation pressures”: what were the applied pressures? And how were these values selected?

On the XRD plots you can start the graphs in 2theta = 30 as there are no peaks below this value.

It would be interest to present the crystal structures of TiO2 and TiO as well as the lattice parameters.

“This section may be divided by subheadings. It should provide a concise and precise description 150 of the experimental results, their interpretation as well as the experimental conclusions that can be 151 drawn.”: remove.

“When sample preparation pressure is too 190 large, the porosity among TiO2 particles is too low”: quantify. What is too large and what is too low? Be specific please.

Can you measure values below 1 % with reliability using SEM? The error would be very large.

Author Response

Thank you for reviewing my manuscript. My reply to your comment is as follows:

Q1: English needs to be revised by a native speaker. Several grammatical mistakes are found in the text.

A1: Thank you for your valuable comments. The language of the full article was revised based on the review opinions of colleagues with experience studying in a country where English is the native language.

Q2: On the importance of Ti and its alloys refer to different potential applications as exemplified in the following papers: 10.1016/j.actamat.2015.12.021, 10.1016/j.matchar.2020.110400 and 10.1016/j.matchar.2020.110180.

A2: Thank you for your valuable comments. I have read the above three articles and have supplemented my introduction based on their content.

Q3: “FFC[2], OS[3], EMR[4], SOM[5, 6].”: write in full the names of these processes.

A3: Thank you for your valuable comments. I have reviewed the full article. The abbreviation that appears for the first time in the article is explained and supplemented. (eg lines 37-38,41,51)

Q4:“Materials and Methods should be described with sufficient details to allow others to replicate 79 and build on published results. Please note that publication of your manuscript implicates that you 80 must make all materials, data, computer code, and protocols associated with the publication available 81 to readers. Please disclose at the submission stage any restrictions on the availability of materials or 82 information. New methods and protocols should be described in detail while well-established 83 methods can be briefly described and appropriately cited.”: this should be removed from the paper.

A4: Thank you for your valuable comments. The relevant paragraph has been deleted.

Q5:What was the grain size of TiO2?

A5:The particle diameter of TiO2 is 100~600nm. The ruler in Figure 1 has an error, which has been corrected and consistent with the text.

Q6: “preparation pressures”: what were the applied pressures? And how were these values selected?

A6: Thank you for your valuable comments. The sample preparation pressure is set by a sample preparation equipment with a digital pressure gauge to ensure that the material is formed while ensuring different relative densities. This can ensure that the porosity of Mg mass transfer is incorrect, and then determine its influence on the reaction process.

Q7: On the XRD plots you can start the graphs in 2theta = 30 as there are no peaks below this value.

A7: Thank you for your valuable comments. The XRD measurement data starts at 10°. I want to show that there are no other phase peaks, especially the TiO2 phase diffraction peak. Figure 2 has been modified to magnify the Mg diffraction peak more clearly.

Q8: It would be interest to present the crystal structures of TiO2 and TiO as well as the lattice parameters.

A8: Thank you for your valuable comments. The TiO2 raw material is a tetragonal rutile structure. TiO and α-Ti are BCC and HCP structures respectively, and their crystal parameters have been supplemented.

Q9: “This section may be divided by subheadings. It should provide a concise and precise description 150 of the experimental results, their interpretation as well as the experimental conclusions that can be 151 drawn.”: remove.

A9: Thank you for your valuable comments. The relevant paragraph has been deleted.

Q10: “When sample preparation pressure is too 190 large, the porosity among TiO2 particles is too low”: quantify. What is too large and what is too low? Be specific please.

A10: Thank you for your valuable comments. It has been explained in the relevant paragraphs. When the sample preparation pressure is 25MPa, the material contact area will be small. Increasing the sample preparation pressure to 150MPa will result in poor Mg mass transfer.

Q11: Can you measure values below 1 % with reliability using SEM? The error would be very large.

A11: Thank you for your valuable comments. The composition results displayed in the SEM are given by the EDS equipment. The results of EDS can reach the percentile, which is used to describe the composition of the particles in the article. Other oxygen content results are macroscopic results measured by an oxygen, nitrogen and hydrogen analyzer.

Reviewer 3 Report

Notes for authors:

  1. The introduction does not consider works on the reduction of TiO2 with magnesium, including in the combustion mode (Bolivar V.Sc.R., Friedrich B. Synthesis of titanium via magnesiothermic reduction of TiO2 (pigment) // Proceedings of EMC .2009; Won C.W., Nersisyan H.H., Won H.I. Chem. Engin. J. 2010. V. 157. P. 270–275; Nersisyan H.H. et al. // Materials Chemistry Physics. 2013. V. 141. P. 283–288; Nersisyan H.H. et al. // Chem. Engin. J. 2014. V. 235. P. 67–74.)
  2. No data on combustion rate versus sample preparation pressure.
  3. Line 137. « Titanium exists in two phases of α-Ti in the form of Ti2O and TiO.». α-Ti is metal, Ti2O and TiO are oxides. These are different phases. α-Ti cannot be in the form of Ti2O and TiO. Same as Line 7,148, 351.
  4. Line 302. «According to the analysis of the reaction process, Mg thermal fluid will reach a new reaction interface by diffusion to initiate a new combustion reaction to achieve the purpose of self-sustaining reaction.» Magnesium heat carrier due to thermal conductivity, not mass transfer.
  5. Line 336. «According to Fig. 5(b) and Fig. 6, this intermediate is a 336 fine pore structure with particle size of 15.1% (sample preparation pressure 75 MPa)» ??

Author Response

Thank you for reviewing my manuscript. My reply to your comment is as follows:

Q1:The introduction does not consider works on the reduction of TiO2 with magnesium, including in the combustion mode (Bolivar V.Sc.R., Friedrich B. Synthesis of titanium via magnesiothermic reduction of TiO2 (pigment) // Proceedings of EMC .2009; Won C.W., Nersisyan H.H., Won H.I. Chem. Engin. J. 2010. V. 157. P. 270–275; Nersisyan H.H. et al. // Materials Chemistry Physics. 2013. V. 141. P. 283–288; Nersisyan H.H. et al. // Chem. Engin. J. 2014. V. 235. P. 67–74.)

A1: Thank you for your valuable comments. I have carefully read the mentioned literature and added it in the introduction.

Q2:No data on combustion rate versus sample preparation pressure.

A2: Thank you for your valuable comments. I have added data on the burning speed of the briquettes with different sample preparation pressures in the article, and analyzed the results in conjunction with the relevant results.

Q3: Line 137. « Titanium exists in two phases of α-Ti in the form of Ti2O and TiO.». α-Ti is metal, Ti2O and TiO are oxides. These are different phases. α-Ti cannot be in the form of Ti2O and TiO. Same as Line 7,148, 351.

A3: Thank you for your valuable comments. I have re-checked the relevant data, corrected the phase, and corrected Ti2O to α-Ti phase.

Q4: Line 302. «According to the analysis of the reaction process, Mg thermal fluid will reach a new reaction interface by diffusion to initiate a new combustion reaction to achieve the purpose of self-sustaining reaction.» Magnesium heat carrier due to thermal conductivity, not mass transfer.

A4: Thank you for your valuable comments. I have carefully reviewed the relevant manuscripts and changed the expression of mass transfer to thermal conductivity.

Q5: Line 336. «According to Fig. 5(b) and Fig. 6, this intermediate is a 336 fine pore structure with particle size of 15.1% (sample preparation pressure 75 MPa)» ??

A5: Thank you for your valuable comments. The particle size in this sentence has been changed to oxygen content.

Reviewer 4 Report

The manuscript submitted by Fan, present the effect of sample preparation pressure on transformation of titanium oxide. The authors present well the subject, but minor amendments must be donned prior to publishing.

Several criticisms are listed below:

Page 2, lines 79 – 84 should be omitted!

From Fig. 1 it seems that the range of TiO2 particles are in micrometric range, please correct.

Is hard to see the Mg peaks in Fig 1. Maybe a zoom on this part could help.

Page 5, lines 150 – 152 should be omitted!

In Table 2, 3 second decimal is not necessary.

Relative density can be given with 2 decimals.

References must be verified for consistency of writing.

Author Response

Thank you for reviewing my manuscript. My reply to your comment is as follows:

Q1:Page 2, lines 79 – 84 should be omitted!

A1: Thank you for your valuable comments. The relevant paragraph has been deleted.

Q2:From Fig. 1 it seems that the range of TiO2 particles are in micrometric range, please correct.

A2:The particle diameter of TiO2 is 100~600nm. The ruler in Figure 1 has an error, which has been corrected and consistent with the text.

Q3: Is hard to see the Mg peaks in Fig 1. Maybe a zoom on this part could help.

A3: Thank you for your valuable comments. The Mg diffraction peak in Figure 2 has been enlarged.

Q4: Page 5, lines 150 – 152 should be omitted!

A4: Thank you for your valuable comments. The relevant paragraph has been deleted.

Q5: In Table 2, 3 second decimal is not necessary.

A5: Thank you for your valuable comments. The number of decimal places in Table 2 has been corrected.

Q6: Relative density can be given with 2 decimals.

Q6: Thank you for your valuable comments. The expression of relative density has been corrected.

Q7: References must be verified for consistency of writing.

A7: The format of the references has been reviewed and revised again, including supplementary references.

Round 2

Reviewer 2 Report

Manuscript was significantly improved and can now be accepted for publication.